# Can the Inclusion of Forage Chicory in the Diet of Lactating Dairy Cattle Alter Milk Production and Milk Fatty Acid Composition? Findings of a Multilevel Meta-Analysis

**DOI:** 10.3390/ani14071002

**Published:** 2024-03-25

**Authors:** Mancoba C. Mangwe, Racheal H. Bryant, Antonia Olszewski, Hitihamy Mudiyanselage Gayani P. Herath, Omar Al-Marashdeh

**Affiliations:** Faculty of Agriculture and Life Sciences, Lincoln University, P.O. Box 85084, Lincoln 7647, New Zealand; racheal.bryant@lincoln.ac.nz (R.H.B.); antonia.olszewski@lincolnuni.ac.nz (A.O.); gayani.herath@lincoln.ac.nz (H.M.G.P.H.); omar.al-marashdeh@lincoln.ac.nz (O.A.-M.)

**Keywords:** *Cichorium intybus* L., *Trifolium repens* L., *Lolium perenne* L., milk production and composition

## Abstract

**Simple Summary:**

Forage chicory is one of the common herbs that has been posited as a complementary species to the traditional ryegrass/white clover mix in pasture-based systems, presenting the benefits of improved mineral nutrition and high nutritive feed during late spring and summer when there is a deficit. This work synthesised data from 15 unique research publications, examining the effect of chicory on milk production and milk fatty acid composition. The results reveal that the effect of chicory on milk production differed as a function of control forage type. Chicory inclusion into the diet of lactating dairy cattle increased milk yield and solids (milk fat + protein) when compared with cows grazing grass-based swards but was similar when compared with cows grazing other forages such as legumes. The increases in milk production when chicory was compared with grasses were associated with concomitant increases in dry matter and metabolisable energy intakes. Moreover, the milk that cows on chicory produced was higher in Omega-3 fatty acids such as alpha linolenic acid, which improve its nutritional quality.

**Abstract:**

In traditional ryegrass/white clover (*Lolium perenne* L./*Trifolium repens* L.) pastoral systems, forage herbs such as chicory (*Cichorium intybus* L.) present an opportunity to fill feed deficits during late spring and summer. Although multiple research publications have evaluated the efficacy of chicory for enhancing milk production and milk fatty acid (FA) profile, no publication has quantitatively synthesised the body of research. This systematic review and meta-analysis examined the effect of chicory on milk production and composition, as well as on the milk fatty acid composition of dairy cattle. A total of 29 comparisons from 15 unique research publications involving 597 dairy cattle were used to develop a dataset for analysis. Three-level random-effect and robust variance estimator models were used to account for the hierarchical structure of the data and the dependency of effect sizes within publications. Chicory inclusion increased milk yield when compared to grass-based diets {weighted mean difference (WMD) = 1.07 (95% CI 0.54–1.60) kg/cow/d, *p* < 0.001}, but it provided a similar milk yield when compared to other forages such as legumes and herbs {dicots; WMD = −0.30, (95% CI −89–0.29) kg/cow/day, *p* = 0.312}. Increases in milk yield were congruent with differences in DM intake (*p* = 0.09) and ME intakes (*p* = 0.003), being similar in chicory-fed and dicot-fed cows but higher than grass-fed cows. Chicory feeding’s effect on milk solids was twice as high during mid lactation {154 days in milk; WMD = 0.13, (95% 0.081–0.175) kg/cow/day, *p* < 0.001} as during late lactation {219 days in milk; WMD = 0.06, (95% 0.003–0.13) kg/cow/day, *p* = 0.041}. In line with milk yield, greater and more significant effect sizes were found for alpha linolenic acid {ALA; WMD = 0.20 (95% CI 0.06–0.35) g/100 g FA, *p* = 0.011} when chicory was compared to grass species only. Comparing chicory with dicots suggests that chicory inclusion did not impact ALA concentrations {WMD = 0.001 (95% CI −0.02–0.2) g/100 g FA, *p* = 0.99}. There were no differences in conjugated linoleic acid concentration in the milk of cows fed chicory or control diets. The study provides empirical evidence of chicory’s efficacy for improved milk production and milk fatty acid composition.

## 1. Introduction

Perennial ryegrass/white clover (*Lolium perenne* L./*Trifolium repens* L.; PRWC) is the predominant sward in temperate regions, such as those that occur in New Zealand, Ireland and Australia. The PRWC sward is easy to establish, generally high-yielding and tolerant of an extensive range of grazing management. While the role of the PRWC sward remains unequivocal, its growth and nutritive value are challenged under dry and hot conditions in summer [1]. The frequent occurrence of climate-related extreme weather events such as drought or high rainfall exposes farmers to several production risks, which include an increase in pests and diseases, feed deficits and their consequences on animal health and welfare [2]. Future resilience of farms is likely to require more diversity, such as adopting forage herbs with greater drought resistance than the traditional PRWC swards in order to improve production and profit for dairy cattle producers [3,4].

Forage chicory (*Cichorium intybus* L.) is one of the common herbs that has been posited as a complementary species to PRWC in pasture-based systems, presenting the benefits of improved mineral nutrition and high nutritive feed during late spring and summer when there is a deficit [5,6,7]. Interest has also increased in the potential role of chicory in mitigating environmental impacts through reduced nitrogen (N) loss from urine [8,9] and lower methane [10]. Substituting proportions of PRWC herbage with forage chicory has been shown to increase frequency of urination in dairy cattle, diluting the concentration of N in the urine [11]. This, in turn, diminishes N loss, mitigating environmental pollution linked with the traditional PRWC swards [12].

Forage herbs are often included in diverse swards with grasses and/or legumes or offered as spatially adjacent monocultures [13]. Increased milk yields have been observed in dairy cattle [14] and ewes [15] when grazing chicory-based swards relative to those on PRWC swards. Chicory may increase milk production via improved dry matter (DM) intake [6,14] or enhanced feed quality relative to the traditional PRWC in pasture-based systems [8]. Alternatively, the trace amounts of secondary compounds such as condensed tannins detected in chicory forage may increase feed conversion efficiency, enhancing the performance of the animals [16,17]. Nonetheless, there is also evidence to the contrary, where little to no difference was observed in milk production between chicory-fed and PRWC-fed dairy cattle [18,19]. Clear and consistent effects on milk production and therefore economic benefits are necessary for decision making on strategies to integrate the herb into different dairy production systems. Therefore, it is of interest to synthesise data from different dairy systems on the impact of forage chicory on dairy cattle milk production.

Forage species have major influences on milk quality of ruminants, though forages need to be fed in sufficient proportions to alter individual milk fatty acid (FA) composition [20]. With the increase in consumer awareness of the source and quality of food products, as well as the preference for ruminant products of pasture-based operations, alternative forages that enhance milk quality are sought. The main objective of this systematic review and meta-analysis of research publications was to evaluate the impacts of chicory on milk production and individual milk FA composition.

## 2. Materials and Methods

### 2.1. Article Search and Screening

To create a database for this study, firstly, we conducted a systematic search on 5 February 2024 for scientific publications on the Google Scholar, ScienceDirect, PubMed and Scopus databases using the search terms: ((dairy cattle) OR (Dairy Cow) OR (Dairy Cows)) AND (chicory). We considered the first 20 pages (25 hits per page) for appropriateness in Google Scholar. Secondly, we created a connected paper graph using Muir et al. [21] to further explore relevant publications. Thirdly, we checked reference lists from relevant research and review publications for suitable articles. In all searches, the timelines were not restricted.

Identified publications were exported to Mendeley reference manager version 1.19.8 (Elsevier) for duplicate removal and article screening. After duplicate removal, the remaining publications were exported to the systematic review software Rayyan (Rayyan Systems Ltd., https://www.rayyan.ai/) [22], where they underwent a two-step screening by three reviewers to determine whether the retrieved publications met the inclusion criteria. During the screening process, each reviewer was blinded to the other reviewers’ scoring until all publications had been reviewed. The first level of screening was based on title and abstract, and studies had to meet the following criteria:Written in English.Be an experimental research article.Use lactating dairy cattle as the study population.Report on at least one of the primary outcomes: milk production, milk FA composition and milk urea nitrogen (MUN). Publications had to report at least one measure of statistical variance {standard error of the mean (SEM) or standard error of differences (SED) or *p*-value} for these primary outcome variables.

Studies selected for the second level of screening underwent full-text recovery. The studies had to meet the following pre-requisites to be included in the analysis:Examine the effect of chicory on the primary outcome variables. Publications or treatments within publications that fed chicory in diverse pastures containing other herbs, such as plantain, were not included, since they have shown similar effects to chicory on the primary outcome variables.Chicory had to be compared with other forages in the publication. The control or comparator forages were either grass species, legumes or dicotyledonous forages. Chicory treatments were either pure chicory pastures or mixed with a grass or legume at any proportion. Publications or chicory treatments with swards, including other herbs such as plantain with similar effects on production parameters as chicory, were omitted in the analysis. Moreover, chicory and control swards needed to be fed fresh, not conserved (hay/silage).

### 2.2. Data Extraction

Quantitative data for all the primary outcomes of interest {milk yield, milk solids (fat + protein; MS), milk protein and fat yields, milk protein and fat percentage, milk fatty acids composition (linoleic acid (LA; C18:2 c9,12), alpha linolenic acid (ALA; C18:3 c9,12,15), conjugated linoleic acid (CLA; 18:2 cis-9, trans-11), saturated FA (SFA), polyunsaturated FA (PUFA)), and MUN} were extracted by two independent reviewers. In addition to the treatment means, the SEMs were extracted and used to weight the treatment means in the analyses. Most publications had reported the SEM or SED. In publications that reported SED, the SED was converted to SEM by using the formula SEM=SED×√2. The SEM was squared to obtain the variance of the calculated effect sizes. We contacted authors of publications where most of the criteria were met but the variance was not given (*n* = 2). If there was no response from the authors (*n* = 1), means for the outcome variables without measures of variation were not included in the meta-analysis.

Data from publications were standardized to similar units. For example, MUN values reported in mmol/L were converted to mg/dL by multiplying by 6. Diet fatty acid composition reported in g/100 g FA were converted to g/kg DM by multiplying with the ether extract content (g/kg DM) reported in the publication. In two publications that reported energy-corrected milk, milk yield was re-calculated according to the formular reported in the publication [23]. However, the other publication [24] did not report the formular, and as such, we used the following formular to convert ECM back to milk yield: ECM (kg/day) = milk yield (kg/day) × (0.38 × milk fat% + 0.24 × milk protein% + 0.17 × milk lactose%)/3.14.

Chemical composition data of forages, including DM content, organic matter (OM), crude protein (CP), neutral detergent fibre (NDF), metabolisable energy (ME) and FA composition, were collected. In publications where ME was not provided, it was imputed based on digestible organic matter in the dry matter (DOMD; [25]) as: ME (MJ/kg DM)=DOMD (g/kg DM)×0.016 or ADF as [26]: ME (MJ/kg DM)=16.2−(ADF;g/kg DM×0.0185). Nutrient intakes (g/day) of dietary components (DM, CP, ME, NDF, LA and ALA) were computed based on apparent dry matter (DM) intake and respective chemical composition of the treatment diets within publications. When available, we also collected information from individual publications on year of publication, country where studies were conducted, days-in-milk, stage of lactation, supplementary usage (kg/cow/day), management system, proportion of chicory in the diet (% of total DM intake) and the forage type used as a control.

### 2.3. Statistical Methods

Descriptive statistics, including the mean, median and range for the chemical composition and nutrient intake data collected in the meta-analysis, were calculated. Differences between means from chicory, grasses and dicots were assessed through linear mixed effect models using the lme4 package, version 1.1-35.1 [27] in R. Forage type was used as fixed effect and publication as random effect. The effect of including chicory in the diet of lactating dairy cattle was assessed using the weighted mean differences (WMDs) between diets with chicory and control diets without chicory. It was apparent that in many publications, chicory was compared with several forages, providing more than one effect size that were dependent on each other. As a result, the WMD mean difference was computed by fitting a 3-level meta-analytic model as described by Assink and Wibbelink [28], using the rma.mv function of the metafor package, version 4.4-0 [29] within the statistical software R, version 4.3.2 [30]. The 3-level random effects model dealt with dependency of publication results, enabling the extraction of multiple effect sizes while accounting for the clustering that existed both within and between publications, thereby maximizing statistical power [28]. Moreover, it was demonstrated during the extraction of the data that, in some publications, data were obtained from the same experimental units over several time points during the experiment. To address the hierarchical structure of the data, the robust variance estimator using a sandwich estimator provided within the clubSandwich package, version 0.5.10 [31] was used.

Forest plots were generated for each outcome variable using the forest function within the metafor package [29]. The presence of extreme outliers in the meta-analysis may increase heterogeneity and/or influence the robustness of conclusions drawn from systematic reviews and meta-analyses [32]. As a result, the presence of outliers was investigated. Outliers were defined as mean difference values more than 3 standard deviations from the overall mean [33]. To minimize their impacts, the mean difference values of these extreme values were replaced with new mean difference values that equalled the highest (or lowest) mean difference that fell within 3 standard deviations [34]. Publication bias was investigated by using Egger’s tests for funnel plot asymmetry, using the regtest function in metafor [29] modified for use in 3-level models via estimating the variance component of each outcome as a model covariate [35].

As there was so much variation in methodology between publications (control forage types, stage of lactation, method of lab analysis, etc.), high within- and between-publication heterogeneity was a likely issue. The 3-level random effects model used in this meta-analysis enabled imputations of total variance explained at each level of hierarchy by calculating a multi-level version of heterogeneity variance (I^2^). In traditional 2-level random effects models, the I^2^ statistic represents variation that is not explained by sampling error (between-publication heterogeneity), with I^2^ values < 25% indicating low heterogeneity, 25–50 indicating moderate heterogeneity and >50 indicating substantial heterogeneity [36]. In 3-level random effect models, this heterogeneity variance (I^2^) is split into two parts [28]: one attributable to true effect size differences within clusters (Level 2—within-publication heterogeneity, i.e., effect of diet treatments on milk production within experiments) and the other to between-cluster variation (Level 3—between-publications heterogeneity). Initially, the I^2^ was calculated from a 3-level random effect model without any moderators. Two separate one-tailed log-likelihood ratio tests were performed to assess whether the within- or between-publication heterogeneity was significant. When heterogeneity was significant, univariable association meta-regressions were performed to assess moderating factors that might have influenced chicory’s effects on the outcome variables.

Several categorical variables were recoded and/or collapsed into fewer categories to be used as moderating factors. First, lactation stage was collapsed into early (<100 days in milk), mid (100–200 days in milk) or late (≥200 days in milk) lactation, since stage of lactation is associated with level of intake and animal production. Secondly, the control forage type to which chicory was compared might have influenced the magnitude and size of the observed effect, owing to variations in biochemical attributes between grasses, legumes and other dicotyledonous plants [37]. As a result, control forage was collapsed into either grass species or others, which included legumes, plantain and other dicotyledonous forages, such as turnips, phacelia, etc. (dicots). For ease of reference, the word “dicots” will be used to represent the other forages in this manuscript. To minimize double counting and the unit of analysis problem [36], the control forage was condensed to either grass species or dicots. Chicory proportion was assessed as a continuous variable for its moderation potential on the outcome variables reported. Chicory proportion was recorded as a proportion of the total DM intake. Other potential moderators assessed in the current meta-analysis included DM, CP, ME, NDF, LA and ALA intakes. To evaluate the potential effects of individual nutrient intakes (DM, CP, ME, NDF intakes) on chicory effects, a mean difference between chicory and control diets was imputed and expressed as a percentage. The mean differences were then evaluated for their modulating effects on chicory using a multi-level model that accounted for the three sources of variability, as with intercept-only models. Following recommendations from the literature [38], the meta-regression analyses were performed when at least 6 publications provided effect size estimates. Moreover, meta-regressions were not performed when an individual stratum had less than two effect sizes in the meta-analysis [39].

As continuous variables, chicory proportion and nutrient intakes were assessed as linear variables and were centred around the mean before the moderation analysis so that the model output intercept was biologically meaningful. For all categorical variables, dummy variables were created. Variables with a *p* value < 0.10 were included into a multiple meta-regression model. An initial multiple meta-regression model was built with all the variables identified at the univariate level included. To check whether the variables that were significantly associated with the effect sizes at the univariable meta-regression were or were not explaining similar variation, the backward selection model fitting approach was applied until the log-likelihood ratio test between two nested models had all remaining variables with a *p*-value of less than 0.1 (*p* < 0.1). The I^2^ was then calculated again from the final mixed-effects meta-regression model, which included all the significant moderators in the multiple meta-regression model. The percentage variation explained at each level was reported, as well as the proportional reduction in unexplained variation with the addition of the moderator variables.

## 3. Results

### 3.1. Characteristics of Selected Studies

The current systematic review identified 540 scientific publications from the online databases (339 from ScienceDirect, 111 from Google Scholar, 57 from Scopus and 33 from PubMed; Figure 1), 41 through the connected papers search (Appendix A) and 2 from a search of the relevant literature. A total of 134 articles were removed as duplicates. As a result, 449 publications had their title and abstracts screened. Thirty-seven (37) publications met the title and abstract screening criteria, and their full text was retrieved. Fifteen publications involving 597 dairy cattle met all inclusion criteria after the second level of screening and were included in the study, providing a total of 29 effect sizes (comparisons). Seven of the fifteen publications were conducted in New Zealand, three in Australia, two in Denmark and one each in France, Switzerland and the USA. A full description of included publications is detailed in Table 1. All publications were in English and were published between 1998 and 2021. The varieties of chicory used in the publications were Choice (*n* = 8), Puna (*n* = 3), Grouse (*n* = 1) and not reported (*n* = 3). Chicory was offered either as a pure sward (*n* = 2) or as mixed white clover (*n* = 1), ryegrass (*n* = 1), ryegrass/white clover (*n* = 10), grasses and legumes (>4 species; *n* = 1). Of the 29 comparisons, chicory was either compared with grass/white clover binary mix (*n* = 19) or diverse swards (*n* = 10). The predominant grass species chicory was compared to was perennial ryegrass. Dicots included legumes {i.e., red clover (*Trifolium pratense* L.), white clover (*Trifolium repens* L.) and lucerne (*Medicago sativa* L.)}, plantain (*Plantago lanceolata* L.), and other dicotyledonous forages such as berseem clover (*Trifolium alexandrinum*), buckwheat (*Fagopyrum esculentum*), phacelia (*Phacelia tanacetifolia*) and turnips (*Brassica rapa*). Experiments were conducted during mid lactation (154 days in milk) in 11 publications and late lactation (218 days in milk) in 4 publications. Dairy cows were supplemented with hay and concentrates during the experiment, with daily supplements ranging from 3.8 to 9.5 kg/cow/day. Eleven of the publications grazed their cattle on pasture the whole day, whilst four confined them indoors the whole day.

### 3.2. Diet Chemical Composition and Nutrient Intakes

The chemical components of the diet of the dairy cattle included varied across publications, depending on the forage type (Table 2). Dry matter content ranged from 81 to 352 g/kg of fresh weight (FW) and was similar between chicory and dicots but 30% less than grass species (*p* < 0.001). There were no differences in NDF content (g/kg DM) between chicory and dicots, but both were, on average, 30% less than grass species. Metabolisable energy was highest in chicory, intermediate in dicots and lowest in grasses (*p* = 0.007). While CP content was variable (ranging between 61 and 265 g/kg DM), it was similar across forages, averaging 179 g/kg DM. The concentrations (g/100 g FA) of palmitic acid (C16:0) and ALA were similar across forages. Linoleic acid concentration was similar between chicory and dicots but nearly 1.9 times greater than that in grass species.

Dry matter intake ranged from 10 to 22.9 kg/cow/day across diets (*p* = 0.094). Estimated ME intake was highest in chicory, intermediate in dicots and lowest in grass-fed cows (*p* = 0.003). Neutral detergent fibre was similar between chicory and dicots but 25% less than that in grass-fed cows. Crude protein intake (1092–4992 g/day) and CP:ME intake ratio (10.5–21.7 g/MJ) varied between publications but were similar amongst the three forages. Linoleic acid intake was similar for chicory- and dicot-fed cows but greater for both than for grass-fed cows. There were no differences in ALA intake between the three forage-based diets (Table 2).

### 3.3. Overall Effect of Chicory

A summary of the overall effect of chicory feeding is presented in Table 3. Each overall effect represents the effect of including the herb chicory relative to the control forages for each of the primary outcome variables reported in the current study. Forest plots displaying the mean and confidence intervals for each study, illustrating the variation between studies, ordered from lowest to greatest values, are presented in Figure 2, Figure 3, Figure 4, Figure 5 and Figure 6.

The overall effect of chicory (Table 3) was significant for milk yield {*p* = 0.024; WMD = 0.624 ± 0.26 kg/cow/day, 95% CI (0.09–1.161), Figure 2}, milk solids {*p* = 0.001; WMD = 0.094 ± 0.02 kg/cow/day, 95% CI (0.054–0.134), Figure 3}, milk fat yield {*p* = 0.003; WMD = 0.104 ± 0.03 kg/cow/day, 95% CI (0.05–0.162)}, LA {*p* = 0.0001; WMD = 0.29 ± 0.048 g/100 g FA, 95% CI (0.21–0.38), Figure 4} and ALA {*p* = 0.033; WMD = 0.154 ± 0.063 g/100 g FA, 95% CI (0.015–0.293), Figure 5}. There was a trend for increased milk protein yield {*p* = 0.055; WMD = 0.077 ± 0.034 kg/cow/day, 95% CI (0.002–0.157)} and PUFA {*p* = 0.067; WMD = 0.555 ± 0.266 g/100 g FA, 95% CI (−0.047–1.157)} when chicory was included in the diet. The overall effect of chicory on milk content of protein, fat, lactose, CLA (Figure 6), saturated, and polyunsaturated FA as well as MUN were not significant, meaning that these outcome variables did not significantly deviate from zero (Table 3).

The findings of the Egger’s regression test for funnel plot asymmetry found no significant publication bias for all the outcome variables (Table 3). Within-publication heterogeneity (Level 2 I^2^; *p* < 0.05) was significant for milk yield, milk solids, milk fat daily yield, LA, ALA, PUFA and MUN concentrations. Between-publication heterogeneity (Level 3 I^2^; *p* < 0.05) was significant for milk protein yield and CLA (Table 3). As a result, moderation analyses were conducted for milk yield, milk solids, LA, ALA and CLA to determine publication attributes that could explain the observed within- and between-publication heterogeneity. Moderation meta-regressions were not conducted for milk protein yield, polyunsaturated FA and MUN because they were reported in less than six unique research publications [38].

### 3.4. Moderator Analyses

An overview of the univariable associations for all the moderating factors investigated are presented in Table 4 and Table 5. No significant moderators emerged for CLA (Table 4), implying that the effect of chicory on CLA was not biased by any of the moderators tested in the current analyses.

#### 3.4.1. Milk Yield

The variables that moderated the effect of chicory on milk yield were diet ME intake, DM intake and forage type (*p* ≤ 0.001; Table 4). Greater effect sizes of chicory inclusion were found when the intakes of ME and DM increased. Moreover, the effect of chicory on milk yield was significant when chicory was compared to grass species {WMD = 1.07, (95% 0.54–1.60) kg/cow/day, *p* < 0.001}. There were no significant differences in effect sizes when chicory was compared against dicots {WMD = −0.296, (95% CI −0.886–1.39) kg/cow/day, *p* = 0.312}. Further multivariate meta-regressions revealed that control forage type and DM intake were the important moderators of chicory’s effect on milk yield, implying that the effect of ME intake at univariable analysis was confounded with the effect of DM intake or control forage type.

Accounting for control forage type and DM intake in the multilevel meta regression model decreased the observed heterogeneity of effect sizes. In the intercept-only models, within- and between-publication heterogeneity accounted for a total of 75% (sum of Level 2 and Level 3 I^2^) of the variation in effect sizes. Subsampling error within experiments accounted for 28.4% of the observed variation. Accounting for both variables in the final model decreased within- and between-publication heterogeneity to 29.3% whilst increasing variation explained by subsampling error to 70.7%.

#### 3.4.2. Milk Solids

The variables that moderated the effect of chicory on milk solids were diet NDF intake, ME intake, control forage type and stage of lactation (*p* ≤ 0.048; Table 4). Greater effect sizes of chicory inclusion were found when the intakes of ME increased. Neutral detergent fibre intake was associated with decreases in effect sizes. The effect of chicory on milk yield was increased when chicory was compared with grasses {WMD = 0.118, (95% 0.08–0.16) kg/cow/day, *p* < 0.001} rather than dicots {WMD = 0.047, (95% CI −0.01–0.10) kg/cow/day, *p* = 0.095}. The effect of including chicory was twice as high during mid lactation {WMD = 0.13, (95% 0.081–0.175) kg/cow/day, *p* < 0.001} as during late lactation {WMD = 0.06, (95% 0.003–0.13) kg/cow/day, *p* = 0.041}. Further multivariate meta-regressions revealed that control forage type, stage of lactation and NDF intake were the major moderators of chicory effect on milk yield, implying that the effect of ME intake in univariable analysis was confounded with either one or all of the final significant factors.

Accounting for control forage type, stage of lactation and NDF intake in the multilevel meta regression model decreased the observed variation in effect sizes. In the intercept-only models, within- and between-publication heterogeneity accounted for a total of 69.5% (sum of Level 2 and Level 3 I^2^) of the variation in effect sizes. Within-experiment subsampling error accounted for 30.5% of the observed variation. Accounting for the three variables in the final multivariable meta-regression model decreased within- and between-publication heterogeneity to 34.8% whilst increasing variation explained by subsampling error to 65.2%.

#### 3.4.3. Individual Milk FA Composition

Dry matter intake was the only significant moderating factor of chicory’s effect on LA (Table 5; *p* = 0.027). Greater and significant effect sizes were found when DM intake increased, such that a 10% increase in DM intake was associated with a 0.077 g/100 g FA increase in LA. Accounting for DM intake in the model decreased within- and between-publication heterogeneity from 91.2% to 76.5%.

Control forage type and DM intake were identified as significant moderators of chicory’s effect on ALA (*p* ≤ 0.084; Table 5). A greater concentration of ALA was observed when chicory was included in the diet relative to grasses {WMD = 0.20 (95% CI 0.06–0.35) g/100 g FA, *p* = 0.011}. There was no significant effect in ALA when chicory was compared with dicots {WMD = 0.001 (95% CI −0.02–0.2) g/100 g FA, *p* = 0.99, Table 5}. As with LA, greater and significant effect sizes were found when DM intake increased, such that a 10% increase in DM intake was associated with a 0.11 g/100 g FA increase in ALA concentration. Accounting for both variables marginally reduced the sum of within and between heterogeneity from 94.4% to 93.3%.

## 4. Discussion

### 4.1. Milk Production and Composition

The results showed that chicory inclusion slightly increased milk yield by 0.62 kg/cow/day compared to control diets. In general, the findings provide empirical evidence of chicory’s efficacy for improved milk production and accord with observations from a previous review of the herb’s effect on animal production [46]. However, moderation analyses indicated that greater and significant effects on milk yield were found only when chicory was compared to grass species. Comparing chicory with dicot-fed cows showed no significant effect (Table 4), meaning that the impact of chicory on milk yield was relatively similar to that of dicots.

The moderation analysis further indicated that high milk yield from chicory-fed cows was mainly driven by increased DM and ME intakes. Dry matter and ME intakes were increased by 0.8 kg/day (*p* = 0.094) and 22 MJ/day (*p* = 0.003), respectively, in cows fed chicory compared with those fed grass-based diets (Table 2). The low fibre content in chicory-based diets might have allowed for increased digestibility and intensity of fermentation in the rumen [47], enhancing DM intake and the supply of nutrients required for milk production. Compared with dicot-based diets, DM, ME and NDF intakes were similar to chicory-based diets, supporting the similarities observed in milk production between the two groups.

Differences in milk solids were greater during mid lactation than late lactation, reflecting the seasonality of the quality of pasture in pastoral systems that are predominant in New Zealand [1]. During mid lactation in summer, a decrease in nutritive value of the traditional grass-based diets (i.e., PRWC) frequently limits milk production [2]. Meanwhile, deep-rooted and heat-tolerant herbs such as chicory and plantain are able to maintain their nutritive value, providing sufficient nutrients to maintain or increase milk solid production [46,48].

Several publications have postulated that substituting grass species with substantial proportions of chicory could reduce milk fat content because of the herb’s minimal fibre contents [15,37]. As such, several researchers allocated chicory along with high-fibre diets to compensate the low fibre contents reported in the herb {e.g., [18,19]. However, at no time in the literature has feeding chicory been associated with reduced dairy cattle milk fat content despite having less NDF content than thresholds (30% of DM) defined for inducing milk fat depression [49]. In a study conducted in Australia, it was reported that although chicory-fed cattle selected diets having <25 NDF%, the resulting milk fat concentration was similar to that of cows on ryegrass with NDF concentrations > 40% [21]. In accordance with the literature, feeding chicory did not reduce milk fat composition relative to control cows in the current study despite the 25% difference (5.5 vs. 7.4 kg; Table 2) in NDF intake between the two diets. Therefore, it appears that regardless of the herb displaying less NDF contents than the recommended 30–40% for diets of lactating dairy cows, feeding the herb at proportions > 40% of DM intake is not associated with milk fat depression. However, our findings should be treated with caution because of the limited number of publications enrolled in the current analysis and the short-duration nature (<30 days) of the publications included. The long-term effects of feeding chicory on rumen functioning and milk fat composition warrant further investigation.

### 4.2. Individual Milk FA Composition

The intake of PUFA such as CLA enhances human health by preventing certain forms of cancer [50,51]. The findings reveal that chicory inclusion maintained milk CLA concentration. The absence of effect of chicory on CLA observed in the present reflects the inconsistencies in the publications included in the analysis. Muir et al. [19] demonstrated a 30% reduction in milk CLA concentrations from chicory-fed cows compared to ryegrass-fed cows. Other publications observed no variation in milk CLA of dairy cattle [23] or ewes [15] offered the herb relative to control animals on grass diets. Muir et al. [21] demonstrated greater (18%) milk CLA concentration for chicory-fed dairy cattle when the herb substituted 25% of ryegrass in the diet. Therefore, based on the combined results of our analysis and the literature, it is not possible to conclude whether chicory inclusion increases or decreases CLA concentration in milk of dairy cattle.

Intake of ALA has been associated with protecting the brain from stroke through exertion of neuroprotection and vasodilation of brain arteries [52]. The current results revealed that milk fatty acids of chicory-fed cattle contained greater levels of ALA relative to control cows. In a confinement study, Kalber et al. [23] evaluated the impact of feeding dicotyledonous plants, e.g., chicory, berseem clover, buckwheat and phacelia, with ryegrass on milk FA composition. The impact of chicory on ALA was similar to that of berseem clover and phacelia but was greater than that of ryegrass [23]. In a related publication that compared chicory to ryegrass and plantain, milk ALA concentrations were higher for chicory-fed cows than ryegrass, while plantain-fed cows showed intermediate concentrations [41]. The general pattern of chicory’s effect on milk FA composition in the analysis aligns with these previous experiments on milk FA composition, as the effect of chicory on ALA was a function of control forage type. When chicory was compared to grass species, the effect size increased significantly. However, the effect of chicory was not significant when chicory was compared to dicots.

The effect of chicory on ALA was moderated by DM intake (*p* = 084), such that increases in DM intake were associated with concomitant increases in concentrations of ALA in the milk of the dairy cows fed. The concentration and estimated intake of ALA from cows fed chicory were similar to those of cows fed grass-based diets. It appears that the elevated ALA content in the milk fat of the chicory-fed cows was mainly realized through increased transfer rate of ALA from feed to milk. Typically, transfer of PUFA from herbage to milk is modest at <10% because of biohydrogenation in the rumen [53]. Postulations based on mean milk fat content (4.5% vs. 4.4%), milk yield (17.7 vs. 16.9 kg/day) and milk ALA content (1.14 vs. 0.74 g/100 g FA) for chicory vs. grass-fed cows, respectively, along with the respective ALA intake estimates reported in Table 2, revealed that the transfer rate of ALA from diet to milk was 4.3% for chicory-fed and 2.9% for grass-fed cows. Compared to this result, the transfer rate of ALA from chicory herbage is greater than the 3.6% reported for grazed dairy cattle in Canterbury, New Zealand [41], but less than the 5.9% reported for confined dairy cattle in Zurich, Switzerland [23]. The increased transfer rate in chicory-fed cows could be a result of greater ruminal passage rate due to the lower fibre and high moisture content of the herb, which reduced exposure of dietary fats to rumen biohydrogenation [54,55]. Alternatively, plant secondary compounds such as the condensed tannins found in chicory [16,37] might have an inhibitory effect on biohydrogenation of PUFA in the rumen, thus increasing their post-ruminal recovery [55]. Kalber et al. [23] speculated that the higher concentration of total phenols in chicory compared to ryegrass herbage (93.1 vs. 66.0 mg/kg) explained the higher transfer rate of ALA from feed to milk. However, it has to be acknowledged that most of the publications included in our analysis did not report the concentration of secondary compounds; thus, it is difficult to make a direct association between milk ALA concentration and specific bioactive compounds found in the herbage.

## 5. Conclusions

Overall, the current analysis revealed that chicory feeding improves milk yield, milk solids, LA and ALA, although further analysis revealed that the magnitude of chicory’s effect on milk production differed as a function of control forage type. Chicory is more efficacious when compared to grass species than it is when compared with other forages such as legumes and herbs. The increases in milk yield were associated with concomitant increases in DM and ME intakes. Increases in milk ALA, on the other hand, were not concomitant with increases in chicory herbage ALA concentrations relative to grass species. The greater ALA content in the milk of cattle fed chicory despite similar diet ALA precursors across forages might be a result of increased transfer rate of ALA from feed to milk due to higher passage rate of digesta in the rumen. Chicory has less fibre but high moisture content, which might have reduced exposure of dietary fats to rumen biohydrogenation.

## Figures and Tables

**Figure 1 animals-14-01002-f001:**
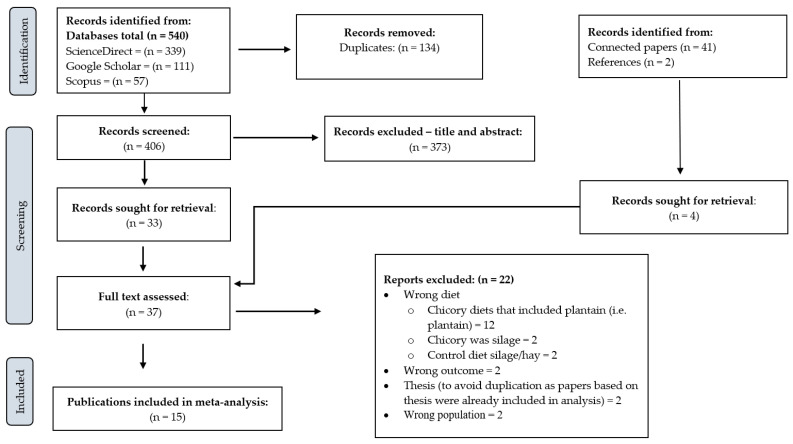
Schematic representation of the selection and screening process of articles included in the meta-analysis.

**Figure 2 animals-14-01002-f002:**
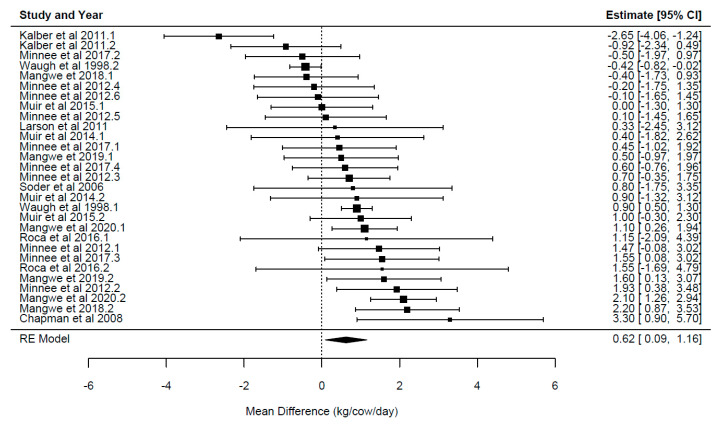
Ordered forest plot of mean difference in milk yield (kg/cow/day) and 95% confidence intervals from 29 observations in 13 publications {Chapman et al. (2008) [14], Kalber et al. (2011) [23], Larsen et al. (2012) [40], Mangwe et al. (2019) [11], Mangwe et al. (2020) [41], Minneé et al. (2017) [8], Mangwe et al. (2020) [42], Muir et al. (2015) [21], Muir et al. (2014) [19], Minneé et al. (2012) [43], Roca-Fernández et al. (2016) [44], Soder et al. (2006) [45], Waugh et al. (1998) [37]} investigating effects of including chicory into the diet of dairy cattle on milk production. The diamond represents the pooled effect mean difference from all studies.

**Figure 3 animals-14-01002-f003:**
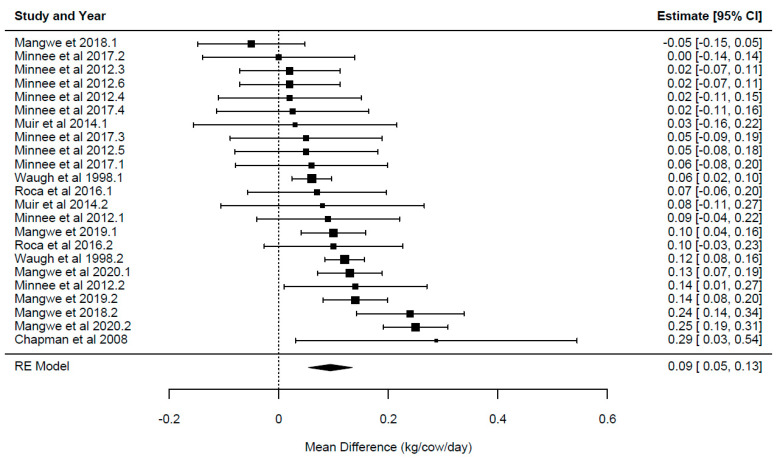
Ordered forest plot of mean difference in milk solids (kg/cow/day) and 95% confidence intervals from 23 observations in 9 publications {Chapman et al. (2008) [14], Mangwe et al. (2019) [11], Mangwe et al. (2020) [41], Minneé et al. (2017) [8], Mangwe et al. (2018) [42], Minneé et al. (2012) [43], Roca-Fernández et al. (2016) [44], Waugh et al. (1998) [37]} investigating effects of including chicory into the diet of dairy cattle on milk solid production. The diamond represents the pooled effect mean difference from all studies.

**Figure 4 animals-14-01002-f004:**
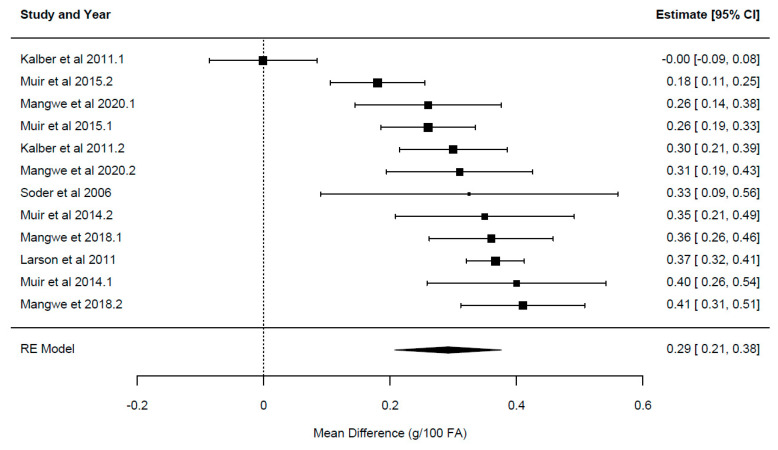
Ordered forest plot of mean difference in linoleic acid [C18:2 c9, 12; g/100 g FA] and 95% confidence intervals from 12 observations in 7 publications {Kalber et al. (2011) [23], Larsen et al. (2012) [40], Mangwe et al. (2020) [41], Mangwe et al. (2018) [42], Muir et al. (2015) [21], Muir et al. (2014) [19], Soder et al. (2006) [45]} investigating effects of including chicory into the diet of dairy cattle on milk fat composition. The diamond represents the pooled effect mean difference from all studies.

**Figure 5 animals-14-01002-f005:**
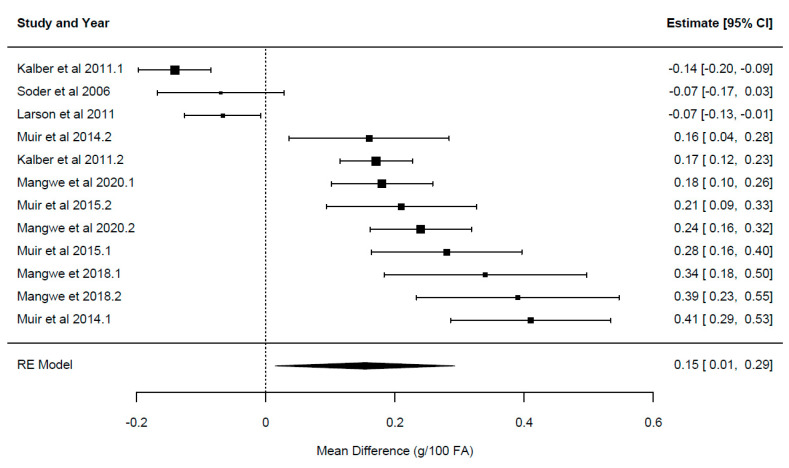
Ordered forest plot of mean difference in alpha linolenic acid [C18:3 c9, 12, 15; g/100 g FA] and 95% confidence intervals from 12 observations in 7 publications {Kalber et al. (2011) [23], Larsen et al. (2012) [40], Mangwe et al. (2020) [41], Mangwe et al. (2018) [42], Muir et al. (2015) [21], Muir et al. (2014) [19], Soder et al. (2006) [45]} investigating effects of including chicory into the diet of dairy cattle on milk fat composition. The diamond represents the pooled effect mean difference from all studies.

**Figure 6 animals-14-01002-f006:**
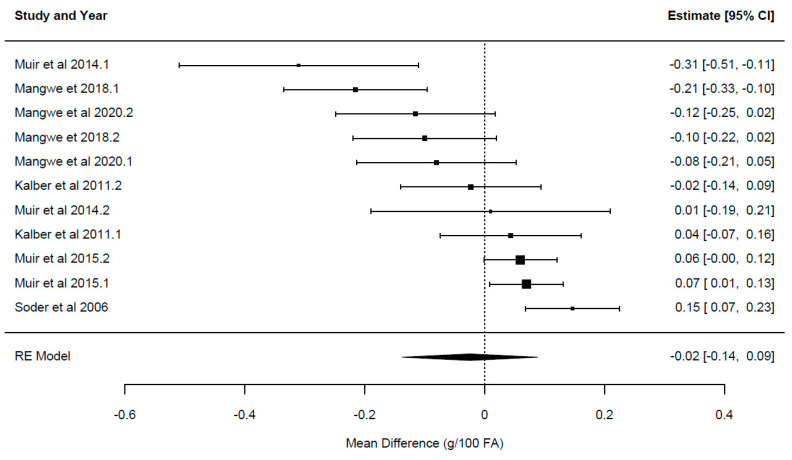
Ordered forest plot of mean difference in conjugated linoleic acid [18:2 cis-9, trans-11; g/100 g FA] and 95% confidence intervals from 11 observations in 6 publications {Kalber et al. (2011) [23], Mangwe et al. (2020) [41], Mangwe et al. (2018) [42], Muir et al. (2015) [21], Muir et al. (2014) [19], Soder et al. (2006) [45]} investigating effects of including chicory into the diet of dairy cattle on milk fat composition. The diamond represents the pooled effect mean difference from all studies.

**Table 1 animals-14-01002-t001:** Description of all 15 publications included in the meta-analysis, including country where the study was conducted, lactation stage, management system, number of treatments (*n*) per publication (studies), interventions and proportion of supplement and chicory (% of total dry matter) in the diet.

Reference	Country	Lactation	System	*n*	Reps	Cattle	Interventions	Chicory Proportion
Andersen et al. (2009) [24]	Denmark	mid	Grazing	6	2	48	Pasture type: white and red clover vs. lucerne vs. chicory	52–72%
Chapman et al. (2008) [14]	Australia	mid	Grazing	5	2	30	Pasture types: grass mixtures vs. chicory	80%
Kalber et al. (2011) [23]	Switzerland	mid	Confined	4	6	28	Pasture types: ryegrass vs. clover vs. phacelia vs. buckwheat vs. chicory	36%
Larsen et al. (2012) [40]	Denmark	mid	Grazing	4	2	48	Pasture type: white clover vs. red clover vs. lucerne vs. mixed pasture vs. chicory	57%
Mangwe et al. (2019) [11]	New Zealand	mid	Grazing	2	3	27	Pasture types: ryegrass vs. chicory vs. plantain	100%
Mangwe et al. (2020) [41]	New Zealand	mid	Grazing	3	3	27	Pasture types: ryegrass vs. chicory	100%
Minneé et al. (2017) [8]	New Zealand	late	Confined	3	6 to 9	42	Pasture types: ryegrass vs. chicory vs. plantain (20 and 40%)	20–40%
Mangwe et al. (2020) [42]	New Zealand	mid	Grazing	3	3	16	Pasture types: ryegrass vs. chicory AM vs. chicory PM	50%
Muir et al. (2015) [21]	Australia	late	Grazing	3	4	36	Pasture types: ryegrass vs. chicory (50 and 100%)	25–50%
Muir et al. (2014) [19]	Australia	mid	Grazing	3	4	72	Pasture types: ryegrass vs. chicory (50 and 100%)	30–60%
Minneé et al. (2012) [43]	New Zealand	late	Confined	10	2	90	Ryegrass vs. chicory vs. plantain (20, 40 and 60%)	20–60%
Mangwe and Bryant (2021) [9]	New Zealand	mid	Grazing	3	3	16	Pasture types: ryegrass vs. chicory AM vs. chicory PM	50%
Roca-Fernández et al. (2016) [44]	France	mid	Grazing	4	4	37	Pasture types: ryegrass vs. pasture mix vs. chicory	30%
Soder et al. (2006) [45]	USA	mid	Grazing	2	5	20	Pasture types: ryegrass vs. chicory	20%
Waugh et al. (1998) [37]	New Zealand	late	Grazing	3	10	60	Pasture types: ryegrass vs. chicory vs. turnips	Not reported

**Table 2 animals-14-01002-t002:** Descriptive statistics of dietary components and nutrient intakes collected during the systematic review, split by forage type (chicory, ryegrass or dicots).

Variable		Overall	Forage Type			*p*-Value
			Chicory	Grasses	Dicots ^1^	
Dry matter (g/kg of FW)	Mean	153	135 ^b^	200 ^a^	140 ^b^	<0.001
	Median	135	119	184	118	
	Range	81–352	81–329	131–352	96–132	
Organic matter (g/kg DM)	Mean	881	865 ^b^	900 ^a^	884 ^ab^	<0.001
	Median	879	868	904	879	
	Range	810–918	810–894	871–918	871–879	
Neutral detergent fibre (NDF: g/kg DM)	Mean	372	324 ^b^	488 ^a^	360 ^b^	<0.001
	Median	377	296	470	302	
	Range	194–635	194–488	366–635	258–678	
Acid detergent fibre (g/kg DM)	Mean	252	226 ^b^	276 ^a^	246 ^ab^	<0.001
	Median	230	225	260	190	
	Range	163–358	163–352	224–358	181–354	
Non-structural carbohydrates (NSC: g/kg DM)	Mean	213	327 ^a^	188 ^b^	306 ^c^	<0.001
	Median	255	401	228	207	
	Range	77–454	89–454	77–261	91–384	
Crude protein (CP: g/kg DM)	Mean	181	175	176	177	0.656
	Median	191	191	190	196	
	Range	61–265	61–249	76–232	129–265	
Metabolisable energy (ME: MJ/kg)	Mean	11.3	11.4 ^a^	10.8 ^b^	11.1 ^ab^	0.007
	Median	11.5	11.4	10.9	11.8	
	Range	7.4–12.9	7.4–12.9	8.2–12.1	9.6–12.9	
CP:NSC (g/g DM)	Mean	1.33	1.09 ^b^	1.57 ^a^	0.56 ^c^	<0.001
	Median	1.07	0.45	1.08	0.5	
	Range	0.28–3.06	0.28–2.46	0.61–3.06	0.27–1.00	
Dry matter intake (kg/day)	Mean	17.2	17.6	16.8	17.3	0.094
	Median	16.1	16.6	15.9	14.9	
	Range	10.6–23.1	13.4–23.1	10.6–22.9	13.6–21.4	
ME intake (MJ/day)	Mean	186	202 ^a^	180 ^b^	193 ^a^	0.003
	Median	182	189	167	172	
	Range	101–298	154–297	101–246	146–276	
NDF intake (g/day)	Mean	6309	5523 ^b^	7585 ^a^	5986 ^b^	<0.001
	Median	6236	5816	7410	4539	
	Range	3010–8381	3108–7686	6166–8381	3962	
Crude protein intake (g/day)	Mean	3181	3246	3182	3219	0.895
	Median	3044	3119	3055	2789	
	Range	1092–4992	2045–4873	1092–4992	2480–4936	
CP:ME intake (g/MJ)	Mean	16	15.2	16.3	15.6	0.256
	Median	16	15.4	16.5	17.3	
	Range	10.5–21.7	10.5–19.6	10.8–21.0	12.4–21.7	
C16:0 (g/kg DM) Palmitic acid	Mean	4.6	4.9 ^a^	3.9 ^b^	5.1 ^a^	0.006
	Median	4.6	4.6	3.4	5.9	
	Range	1.95–6.5	3.64–6.1	1.95–6.4	5.2–6.5	
C18:2 c9, 12 (g/kg DM) Linoleic acid	Mean	5.8	7.2 ^a^	3.4 ^b^	5.5 ^ab^	0.002
	Median	5.7	7.2	3.3	5.3	
	Range	1.58–9.67	4.1–9.7	1.6–5.7	5.2–6.7	
C18:3 c9, 12, 15 (g/kg DM) Alpha linolenic acid	Mean	13.6	14	13.9	12.8	0.683
	Median	12.2	12.2	11.1	17.4	
	Range	1.53–26.9	3.43–24.2	1.53–26.4	9.8	
C18:2 c9, 12 intake (g/day) Linoleic acid	Mean	103.8	128.1 ^a^	71.2 ^b^	112 ^ab^	0.009
	Median	105.5	111.1	68.6	112	
	Range	28.2–186	82.0–185.6	28.3–107	107	
C18:3 c9, 12, 15 intake (g/day) Alpha linolenic acid	Mean	228	231	218	235	0.448
	Median	181	188.7	159	372	
	Range	27–470	66–433	27–411	141–471	

^a–c^ Means of forages within a row with different superscripts differ (*p* < 0.05). ^1^ Includes legumes {i.e., red clover (*Trifolium pratense* L.), white clover (*Trifolium repens* L.) and lucerne (*Medicago sativa* L.)}, plantain (*Plantago lanceolata* L.) and other dicotyledonous forages such as berseem clover (*Trifolium alexandrinum*), buckwheat (*Fagopyrum esculentum*), phacelia (*Phacelia tanacetifolia*) and turnips (*Brassica rapa*).

**Table 3 animals-14-01002-t003:** Weighted mean differences of dry matter intake, milk production, milk composition and functional milk fatty acids (g/100 g of FA) of dairy cattle offered chicory-based diets.

Outcome	Publications	ES ^1^	Effect Sizes (WMD)	95% CI	*p*-Value	%Var. at Level 1 (I^2^)	Level 2 Variance	%Var. at Level 2 (I^2^)	Level 3 Variance	%Var. at Level 3 (I^2^)	Egger’s
Milk yield (kg/cow)	13	29	0.624 ± 0.26	(0.09–1.161)	0.024	28.4	0.527 **	45.6	0.300	25.9	0.142
Milk solids (kg/day)	9	23	0.09 ± 0.02	(0.05–0.13)	0.001	30.6	0.003 *	54.8	0.001	14.7	0.114
Milk protein (kg/day)	5	9	0.077 ± 0.03	(0.002–0.16)	0.055	4.9	0.000	30.3	0.004 **	64.8	0.932
Milk fat yield (g/day)	5	9	0.100 ± 0.03	(0.05–0.16)	0.003	17.3	0.004 **	82.7	0.000	0.00	0.332
Milk protein (%)	11	22	0.02 ± 0.02	(−0.23–0.07)	0.332	57.0	0.002	10.5	0.001	32.5	0.936
Milk fat (%)	11	22	0.077 ± 0.03	(−0.01–0.15)	0.143	86.5	0.003	13.5	0.000	0.0	0.150
Milk lactose (%)	5	11	0.01 ± 0.02	(−0.03–0.04)	0.762	66.9	0.000	0.0	0.001	33.1	0.750
C18:2 c9, 12	7	12	0.29 ± 0.05	(0.21–0.38)	<0.001	8.8	0.019 **	77.4	0.003	13.8	0.162
C18:3 c9, 12, 15	7	12	0.154 ± 0.06	(0.02–0.29)	0.033	5.5	0.016 **	49.6	0.017	45.0	0.280
18:2 cis-9, trans-11	7	12	−0.024 ± 0.05	(−0.14–0.09)	0.277	16.7	0.000	0.00	0.013 **	83.7	0.421
Saturated FA	5	9	−0.396 ± 0.40	(−1.3–0.50)	0.350	57.7	0.126	10.5	0.384	31.8	0.761
Monounsaturated FA	5	9	−0.07 ± 0.66	(−1.59–1.45)	0.921	48.4	0.070	2.80	1.211	48.7	0.887
Polyunsaturated FA	5	9	0.555 ± 0.27	(−0.05–1.16)	0.067	2.40	0.431 *	75.7	0.125	21.9	0.975
Milk urea nitrogen (mg/dL)	5	9	−1.64 ± 0.83	(−3.90–0.63)	0.134	0.00	8.62 ***	99.9	0.000	0.0	0.237

^1^ ES = number of effect sizes; WMD = weighted mean difference, CI = confidence interval. % Var (I^2^) = percentage of variance explained; Level 2 variance = variance estimate between effect sizes from the same publication; Level 3 variance = variance estimate between effect sizes between publications. * *p* < 0.05. ** *p* < 0.01., *** *p* < 0.001.

**Table 4 animals-14-01002-t004:** Meta-regression analysis with potential moderators on dry matter (DM intake) intake and milk production (kg/day) (univariable models).

Outcome	Moderator	*n* ^1^	ES ^1^	Intercept ^1^	95% CI	Coefficient	95% CI	*p*-Value	F (df1, df2)	σ^2^_1_	σ^2^_2_	1I^2^	2I^2^	3I^2^
Milk yield	Chicory proportion	13	29	0.59 *	(0.01–1.09)	0.011	(−0.010–0.030)	0.211	F(1, 27) = 1.64	0.565	0.189	30.4	52.2	17.4
	NDF intake	12	27	0.68 *	(0.03–1.33)	−0.011	(−0.040–0.002)	0.399	F(1, 25) = 0.69	0.579	0.362	36.9	24.3	38.8
	CP intake	13	29	0.70 *	(0.08–1.32)	0.022	(−0.020–0.060)	0.288	F(1, 26) = 1.18	0.445	0.464	37.7	31.8	30.5
	ME intake	11	26	0.81 *	(0.14–1.48)	0.079 **	(0.040–0.120)	0.001	F(1, 24) = 14.4	0.011	0.735	41.7	0.88	57.4
	DM intake	12	27	0.37 **	(−0.12–0.87)	0.091 **	(0.050–0.130)	<0.001	F(1, 25) = 18.3	0.223	0.125	61.2	13.9	24.9
	Control forage type	13	29					<0.001	F(1, 27) = 38.8	0.000	0.436	43.0	0.0	57.0
	Dicots	7	10	−0.30	(−0.89–0.29)									
	Grass	11	19	1.07 ***	(0.54–1.60)	−1.390	(−1.82–−0.92)							
	Lactation	13	28					0.641	F(1, 27) = 0.22	0.504	0.417	26.3	40.3	33.4
	Mid	9	15	0.75	(−0.03–1.53)									
	Late	4	14	0.49	(−0.37–1.34)	−0.266	(−1.420–0.890)							
Milk solids	Chicory proportion	9	23	0.10 ***	(0.05–0.14)	−0.001	(−0.002–0.000)	0.135	F(1, 21) = 2.41	0.002	0.002	28.9	35.8	35.3
	NDF intake	8	21	0.09 ***	(0.05–0.14)	−0.002 *	(−0.004-−0.000)	0.045	F(1, 19) = 4.61	0.002	0.001	45.0	36.0	19.0
	CP intake	8	21	0.10	(0.05–0.15)	0.001	(−0.001–0.003)	0.595	F(1, 19) = 0.29	0.002	0.004	32.7	47.0	20.2
	ME intake	8	21	0.11 ***	(0.05–0.16)	0.004 *	(0.000–0.008)	0.034	F(1, 19) = 5.24	0.004	0.002	30.9	20.0	49.0
	DM intake	8	21	0.10 ***	(0.05–0.15)	0.004	(−0.001–0.006)	0.103	F(1, 19) = 2.93	0.003	0.002	34.5	36.4	29.1
	Control forage type	9	23					0.048	F(1, 21) = 4.40	0.003	0.000	34.2	64.9	0.90
	Dicots	5	8	0.05	(−0.00–0.10)									
	Grass	9	15	0.12 ***	0.08–0.16)	0.071 *	(0.006–0.142)							
	Lactation	9	23					0.041	F(1, 21) = 4.76	0.003	0.000	37.2	62.8	0.00
	Mid	6	11	0.13 ***	(0.08–0.18)									
	Late	3	12	0.06 *	(0.01–0.10)	−0.069	(−0.014–0.003)							

^1^ *n* = number of publications; ES = number of effect sizes; Intercept = mean difference; CI = confidence intervals; Coefficient (β) = estimated regression coefficient; *p* value = of omnibus test; σ^2^_1_ = variance within publications; σ^2^_2_ = variance between publications. 1I^2^ = Level 1 variance (variance estimate attributed to sampling error of the individual study, i.e., the animal level), 2I^2^ = Level 2 variance (variance estimate between effect sizes from the same publication) and 3I^2^ = Level 3 variance (variance estimate between effect sizes between publications). * *p* < 0.05, ** *p* < 0.01, *** *p* < 0.001.

**Table 5 animals-14-01002-t005:** Meta-regression analysis with potential moderators on milk fatty acid (g/100 g FA) (univariable models).

Outcome	Moderator ^1^	*n* ^1^	ES ^1^	Intercept ^1^	95% CI	Coefficient	95% CI	*p*-Value	F (df1, df2)	σ^2^_1_	σ^2^_2_	1I^2^	2I^2^	3I^2^
Linoleic acid	Chicory proportion	7	12	0.29 *	(0.21–0.38)	0.0003	(−0.003–0.004)	0.813	F(1, 10) = 0.06	0.011	0.002	12.4	70.7	15.1
	NDF intake	7	12	0.29 ***	(0.120–0.38)	−0.001	(−0.007–0.005)	0.666	F(1, 10) = 0.20	0.009	0.005	13.8	57.1	29.1
	CP intake	6	11	0.29 ***	(0.19–0.38)	−0.001	(−0.009–0.007)	0.763	F(1, 9) = 0.10	0.010	0.003	12.4	66.4	21.2
	ME intake	6	11	0.29 ***	(0.20–0.37)	0.009	(−0.003–0.022)	0.124	F(1, 9) = 2.87	0.007	0.003	16.1	55.7	28.2
	DM intake	7	12	0.29 ***	(0.22–0.36)	0.008 *	0.001–0.014)	0.027	F(1, 10) = 6.69	0.005	0.002	23.5	58.7	17.7
	LA intake	5	10	0.28 ***	(0.16–0.40)	0.0001	(−0.001–0.001)	0.814	F(1, 8) = 0.60	0.009	0.007	12.7	49.0	38.2
	ALA intake	6	11	0.29 ***	(0.19–0.39)	−0.0003	(−0.002–0.001)	0.625	F(1, 9) = 0.26	0.009	0.005	12.1	57.5	30.4
	Control forage type	7	12					0.137	F(1, 10) = 2.61	0.005	0.009	13.5	30.4	56.0
	Dicots	3	3	0.22 **	(0.07–0.36)									
	Grass	6	9	0.33 ***	0.22–0.44)	0.112	(0.072–0.360)							
	Lactation	7	12					0.272	F(1, 10) = 1.35	0.009	0.002	16.1	66.7	17.2
	Mid	6	10	0.27 ***	(0.18–0.36)									
	Late	1	2	0.39	(0.19–0.58)	0.113	(−0.100–0.330)							
Alpha linolenic acid	Chicory proportion	7	12	0.16 *	(0.02–0.30)	0.001	(−0.004–0.005)	0.683	F(1, 10) = 0.18	0.021	0.012	5.4	59.7	34.9
	NDF intake	7	12	0.13	(−0.01–0.28)	−0.007	(−0.015–0.002)	0.131	F(1, 10) = 2.71	0.012	0.021	5.5	33.9	60.6
	CP intake	6	11	0.19 *	(0.05–0.32)	0.003	(−0.008–0.014)	0.588	F(1, 9) = 0.32	0.023	0.007	5.9	71.7	22.5
	ME intake	6	11	0.18 *	(0.03–0.33)	0.01	(−0.009–0.03)	0.259	F(1, 9) = 1.45	0.015	0.015	5.8	46.7	47.4
	DM intake	7	12	0.15 *	(0.01–0.29)	0.011	(−0.002–0.024)	0.084	F(1, 10) = 3.66	0.011	0.018	6.1	35.3	58.6
	LA intake	5	10	0.23 *	(0.07–0.37)	−0.0002	(−0.001–0.001)	0.541	F(1, 8) = 0.41	0.016	0.012	7.1	52.3	40.6
	ALA intake	6	11	0.15 *	(0.03–0.34)	0.011	(−0.002–0.003)	0.673	F(1, 9) = 0.19	0.000	0.183	5.2	49.0	45.8
	Control forage type	7	12					0.050	F(1, 10) = 4.95	0.009	0.018	6.7	30.2	63.2
	Dicots	3	3	0.001	(−0.20–0.20)									
	Grass	6	9	0.20 *	(0.06–0.35)	0.200	(−0.000–0.4040)							
	Lactation	7	12					0.168	F(1, 10) = 2.20	0.017	0.012	6.3	55.3	38.4
	Mid	6	10	0.12	(−0.02–0.26)									
	Late	1	2	0.37 **	(0.02–0.71)	0.250	(−0.120–0.610)							
Conjugated linoleic acid	Chicory proportion	6	11	−0.027	(−0.13–0.07)	−0.001	(−0.003–0.000)	0.120	F(1, 9) = 2.94	0.000	0.010	20.4	2.1	77.6
	NDF intake	6	11	−0.008	(−0.11–0.09)	0.004	(−0.002–000)	0.172	F(1, 9) = 2.20	0.003	0.006	21.7	27.5	50.7
	CP intake	6	11	−0.056	(−0.16–0.04)	0.003	(−0.002–0.008)	0.165	F(1, 8) = 2.33	0.000	0.007	28.4	0.0	71.6
	ME intake	6	11	−0.058	(−0.17–0.05)	0.001	(0.009–0.012)	0.767	F(1, 8) = 0.09	0.000	0.010	22.9	0.0	77.1
	DM intake	6	11	−0.026	(−0.14–0.09)	−0.001	(−0.010–0.008)	0.798	F(1, 9) = 0.07	0.000	0.014	15.7	0.0	84.3
	LA intake	5	10	−0.060	(−0.15–0.03)	0.0003	(−0.000–0.001)	0.194	F(1, 8) = 2.01	0.000	0.006	32.1	0.0	67.9
	ALA intake	5	10	−0.061	(−0.15–0.02)	0.001	(−0.000–0.001)	0.108	F(1, 8) = 3.27	0.001	0.004	37.7	6.4	55.8
	Control forage type	6	11					0.594	F(1, 9) = 0.31	0.000	0.013	17.2	0.0	82.8
	Dicots	2	2	−0.05	(−0.21–0.11)									
	Grass	4	9	−0.02	(−0.13–0.10)	0.032	(−0.100–0.160)							
	Lactation	6	11					0.228	F(1, 9) = 1.68	0.000	0.011	18.5	0.0	80.5
	Mid	5	9	0.004	(−0.11–0.12)									
	Late	1	2	−0.160	(−0.42–0.10)	−0.161	(−0.440–0.120)							

^1^ *n* = number of publications; ES = number of effect sizes; Intercept = mean difference; CI = confidence intervals; Coefficient (β) = estimated regression coefficient; *p* value = of omnibus test; σ^2^_1_ = variance within publications; σ^2^_2_ = variance between publications. 1I^2^ = Level 1 variance (variance estimate attributed to sampling error of the individual study, i.e., the animal level), 2I^2^ = Level 2 variance (variance estimate between effect sizes from the same publication) and 3I^2^ = Level 3 variance (variance estimate between effect sizes between publications); NDF = neutral detergent fibre, ME = metabolisable energy, CP = crude protein, LA = linoleic acid, ALA = alpha linolenic acid. * *p* < 0.05, ** *p* < 0.01, *** *p* < 0.001.

## Data Availability

Data used to perform analysis are available as Appendix A.

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
