# Peer review of "Can the Inclusion of Forage Chicory in the Diet of Lactating Dairy Cattle Alter Milk Production and Milk Fatty Acid Composition? Findings of a Multilevel Meta-Analysis"

_animals, 2024, doi:10.3390/ani14071002_

Round 1

Reviewer 1 Report

Comments and Suggestions for Authors

Dear Authors,

Your manuscript titled "Can the inclusion of forage chicory in a diet of lactating dairy cattle alter milk production and milk fatty acid composition? Findings of a multilevel meta-analysis" might be suitable for publication at Animals after minor revision. Please, see below a list of comments/suggestions to be addressed by you before accepting it.

L259-L260 Use scientific nomenclature to cite all these species.

L307-L309 Use scientific nomenclature to cite all these species.

L458 Replace 'meta-analysis?' by 'meta-analysis'.

L554 Replace 'function control forage type' by 'function of forage type'.

L577-L714 Check all references according to Animals' instructions for authors (specially the ones with the numbers 8, 44 and 49).

Please, also check all spelling errors throughout your manuscript and correct them (i.e., replace 'pub-lications' by 'pu-blications', 'leg-umes' by 'le-gumes').

Yours sincerely,

Reviewer.

Author Response

Your manuscript titled "Can the inclusion of forage chicory in a diet of lactating dairy cattle alter milk production and milk fatty acid composition? Findings of a multilevel meta-analysis" might be suitable for publication at Animals after minor revision. Please, see below a list of comments/suggestions to be addressed by you before accepting it.

L259-L260 Use scientific nomenclature to cite all these species –

  • We have now included scientific nomenclature for white clover (Trifolium repens) and plantain (Plantago lanceolata L.). Changes are highlighted in yellow L268-271.

L307-L309 Use scientific nomenclature to cite all these species 

  • We have now included scientific nomenclature for all species – “legumes {i.e. red clover (Trifolium pratense), white clover (Trifolium repens L.) and lucerne (Medicago sativa L.)}, plantain (Plantago lanceolata L.), and other dicotyledonous forages such as berseem clover (Trifolium alexandrinum), buckwheat (Fagopyrum esculentum), phacelia (Phacelia tanacetifolia) and turnips (Brassica rapa). Changes are highlighted in yellow L315-318

L458 Replace 'meta-analysis?' by 'meta-analysis'

  • Replaced L462

L554 Replace 'function control forage type' by 'function of forage type'.

  • Replaced L560

L577-L714 Check all references according to Animals' instructions for authors (specially the ones with the numbers 8, 44 and 49).

  • All references are now written according to the Journals’ instructions for authors, thank you.

Please, also check all spelling errors throughout your manuscript and correct them (i.e., replace 'pub-lications' by 'pu-blications', 'leg-umes' by 'le-gumes')

  • Checked and changed, thank you.

Reviewer 2 Report

Comments and Suggestions for Authors

Dear authors, 

The beauty of systematic reviews is that they bring together a number of different work focused on a specific topic. You did a great job to dig information about the use of chicory. You were able to demonstrate that incorporating chicory into the diet of dairy cows enhances milk production, milk solids, linoleic acid (LA), and alpha-linolenic acid (ALA). However, the extent of chicory's impact on milk output varied depending on the type of forage used as a control. Comparatively, chicory proved more effective when compared to grass species rather than other forages like legumes and herbs.

It is a nice read. My only suggestion is to cut down some of the text where possible. The word "meta-analysis" has been used 33 times throughout your manuscript. Perhaps use alternatives such as "current work", "this study", or not even say anything since it is implicit you are discussing the results from your meta-analysis. By the way, in line 48 there is a question mark after "The meta-analysis?" maybe a typo?

Sincerely, 

Reviewer

Author Response

The beauty of systematic reviews is that they bring together a number of different work focused on a specific topic. You did a great job to dig information about the use of chicory. You were able to demonstrate that incorporating chicory into the diet of dairy cows enhances milk production, milk solids, linoleic acid (LA), and alpha-linolenic acid (ALA). However, the extent of chicory's impact on milk output varied depending on the type of forage used as a control. Comparatively, chicory proved more effective when compared to grass species rather than other forages like legumes and herbs.

It is a nice read. My only suggestion is to cut down some of the text where possible. The word "meta-analysis" has been used 33 times throughout your manuscript. Perhaps use alternatives such as "current work", "this study", or not even say anything since it is implicit you are discussing the results from your meta-analysis. By the way, in line 48 there is a question mark after "The meta-analysis?" maybe a typo?

  • Thank you for the compliments. We have minimized the extensive use of the meta-analysis as per the suggestion. We have also corrected the typo on meta-analysis.

Reviewer 3 Report

Comments and Suggestions for Authors

This manuscript reports the results of a meta-analysis on studies utlizing chicory as an alternative to other pasture swards, primarily perennial ryegrass/white clover pasture typically utlized in temperate dairy regions of the world. this research hgihlightd that chicory is a rather undertudued forage source and revealed that in general, despite the forage it was compared to, there was a slight improvement in milk yield and some milk fatty acids, though no improvement in others. 

There are a few minor suggested edits:

Line 58: replace "compared to" with "than".

in paragraph two, lines 60-77, would like more emphasis on why chicory may lead to reduced N loss in Urine and less methane, and that there may be links to secondary compounds for these attributes, as I think that is important to highlight.

Line 107- the study population

line 136- data were standardized to the similar units- I am unclear about what units, please clarify the meaning here.

line 241- go from 37 studies to 15- was a step missing? What criteria eliminated the other 22 studies? would their inclusion have affected the outcomes? Just need more information about why they were taken out.

paragraph of line 267 to 277- change "lower "when used to "less". Since you are reporting numerical values, more correct to use a mathematical term "less" than to use "lower" and throughout the paper.

Line 343- needs a reference about recommendations from previous meta analyses. Not really sure what is meant here.

Ordered forest plots are interesting- maybe more about how they were created in the methods or in the figure description.

Line 461- delete Though- fragment sentence.

In the paragraph line463 maybe add that the difference in milk yiled is not biologically or economically significant.

Line 546- from not form. 

Author Response

This manuscript reports the results of a meta-analysis on studies utlizing chicory as an alternative to other pasture swards, primarily perennial ryegrass/white clover pasture typically utlized in temperate dairy regions of the world. this research hgihlightd that chicory is a rather understudied forage source and revealed that in general, despite the forage it was compared to, there was a slight improvement in milk yield and some milk fatty acids, though no improvement in others. 

Line 58: replace "compared to" with "than".

  • Changes highlighted in yellow L58

in paragraph two, lines 60-77, would like more emphasis on why chicory may lead to reduced N loss in Urine and less methane, and that there may be links to secondary compounds for these attributes, as I think that is important to highlight.

  • Initially, we though probably only need a one sentence statement pointing out the potential benefits for mitigating environmental impacts through reduced N loss from urine and lower methane as any further description may get readers on the environmental bandwagon which is not the focus of this paper. We have now emphasized role of the herb on reduced N. Added information is highlighted in yellow L65-68

Line 107- the study population

  • Changed, L110

line 136- data were standardized to the similar units- I am unclear about what units, please clarify the meaning here.

  • Some outcome variables were reported in different units by different publications. So, we had to convert them to similar units. For example, MUN values reported in mmol/L were converted to mg/dL by multiplying by 6. Diet fatty acid composition reported in g/100 g FA were converted to g/kg DM by multiplying with the ether extract content (g/kg DM) reported in the publication. We have now added that information after the sentence. L144-147

line 241- go from 37 studies to 15- was a step missing? What criteria eliminated the other 22 studies? would their inclusion have affected the outcomes? Just need more information about why they were taken out.

There were two levels of screening. The first level was based on title and abstract whilst the second level of screening was based on full text. 37 articles met the inclusion criteria during the first level of screening. We retrieved full text for those articles, which then underwent the second level of screening. That is where we remained with the fifteen articles. We have now included the details in section 2.1 of the materials and method section L106-127 and section 3.1 of the results section L255-258.

paragraph of line 267 to 277- change "lower "when used to "less". Since you are reporting numerical values, more correct to use a mathematical term "less" than to use "lower" and throughout the paper.

  • We have now replaced lower with less where relevant throughout manuscript.

Line 343- needs a reference about recommendations from previous meta analyses. Not really sure what is meant here.

  • Previous meta-analyses recommended that moderation analysis be conducted when at least 6 publications provided effect size estimates. We have now included the reference in the result (L353) and material sections (L231).

Ordered forest plots are interesting- maybe more about how they were created in the methods or in the figure description.

  • Thank you. Forest plots were generated for each outcome variable using the forest function within the metafor package [27]. Line 184-185

Line 461- delete Though- fragment sentence.

  • Replaced with However, L465

In the paragraph line463 maybe add that the difference in milk yiled is not biologically or economically significant.

  • We have modified the sentence to reflect the modest increase in milk yield “The meta-analysis showed that chicory inclusion slightly increased milk yield by 0.62 kg/cow/day compared to control diets, L462

Line 546- from not form. 

  • Changed, L551